# LightRNN: Memory and Computation-Efficient Recurrent Neural Networks

**Xiang Li**[1]    **Tao Qin**[2]    **Jian Yang**[1]    **Tie-Yan Liu**[2]

[1]Nanjing University of Science and Technology [2]Microsoft Research Asia

[1]`implusdream@gmail.com` [1]`csjyang@njust.edu.cn`

[2]`{taoqin, tie-yan.liu}@microsoft.com`

## Abstract

Recurrent neural networks (RNNs) have achieved state-of-the-art performances in many natural language processing tasks, such as language modeling and machine translation. However, when the vocabulary is large, the RNN model will become very big (e.g., possibly beyond the memory capacity of a GPU device) and its training will become very inefficient. In this work, we propose a novel technique to tackle this challenge. The key idea is to use 2-Component (2C) shared embedding for word representations. We allocate every word in the vocabulary into a table, each row of which is associated with a vector, and each column associated with another vector. Depending on its position in the table, a word is jointly represented by two components: a row vector and a column vector. Since the words in the same row share the row vector and the words in the same column share the column vector, we only need $2\sqrt{|V|}$ vectors to represent a vocabulary of $|V|$ unique words, which are far less than the $|V|$ vectors required by existing approaches. Based on the 2-Component shared embedding, we design a new RNN algorithm and evaluate it using the language modeling task on several benchmark datasets. The results show that our algorithm significantly reduces the model size and speeds up the training process, without sacrifice of accuracy (it achieves similar, if not better, perplexity as compared to state-of-the-art language models). Remarkably, on the One-Billion-Word benchmark Dataset, our algorithm achieves comparable perplexity to previous language models, whilst reducing the model size by a factor of 40-100, and speeding up the training process by a factor of 2. We name our proposed algorithm *LightRNN* to reflect its very small model size and very high training speed.

## 1   Introduction

Recently recurrent neural networks (RNNs) have been used in many natural language processing (NLP) tasks, such as language modeling [14], machine translation [23], sentiment analysis [24], and question answering [26]. A popular RNN architecture is long short-term memory (LSTM) [8, 11, 22], which can model long-term dependence and resolve the gradient-vanishing problem by using memory cells and gating functions. With these elements, LSTM RNNs have achieved state-of-the-art performance in several NLP tasks, although almost learning from scratch.

While RNNs are becoming increasingly popular, they have a known limitation: when applied to textual corpora with large vocabularies, the size of the model will become very big. For instance, when using RNNs for language modeling, a word is first mapped from a one-hot vector (whose dimension is equal to the size of the vocabulary) to an embedding vector by an input-embedding matrix. Then, to predict the probability of the next word, the top hidden layer is projected by an output-embedding matrix onto a probability distribution over all the words in the vocabulary. When

the vocabulary contains tens of millions of unique words, which is very common in Web corpora, the two embedding matrices will contain tens of billions of elements, making the RNN model too big to fit into the memory of GPU devices. Take the ClueWeb dataset [19] as an example, whose vocabulary contains over 10M words. If the embedding vectors are of 1024 dimensions and each dimension is represented by a 32-bit floating point, the size of the input-embedding matrix will be around 40GB. Further considering the output-embedding matrix and those weights between hidden layers, the RNN model will be larger than 80GB, which is far beyond the capability of the best GPU devices on the market [2]. Even if the memory constraint is not a problem, the computational complexity for training such a big model will also be too high to afford. In RNN language models, the most time-consuming operation is to calculate a probability distribution over all the words in the vocabulary, which requires the multiplication of the output-embedding matrix and the hidden state at each position of a sequence. According to simple calculations, we can get that it will take tens of years for the best single GPU today to finish the training of a language model on the ClueWeb dataset. Furthermore, in addition to the challenges during the training phase, even if we can successfully train such a big model, it is almost impossible to host it in mobile devices for efficient inferences.

To address the above challenges, in this work, we propose to use 2-Component (2C) shared embedding for word representations in RNNs. We allocate all the words in the vocabulary into a table, each row of which is associated with a vector, and each column associated with another vector. Then we use two components to represent a word depending on its position in the table: the corresponding row vector and column vector. Since the words in the same row share the row vector and the words in the same column share the column vector, we only need $2\sqrt{|V|}$ vectors to represent a vocabulary with $|V|$ unique words, and thus greatly reduce the model size as compared with the vanilla approach that needs $|V|$ unique vectors. In the meanwhile, due to the reduced model size, the training of the RNN model can also significantly speed up. We therefore call our proposed new algorithm *(LightRNN)*, to reflect its very small model size and very high training speed.

A central technical challenge of our approach is how to appropriately allocate the words into the table. To this end, we propose a bootstrap framework: (1) We first randomly initialize the word allocation and then train the LightRNN model. (2) We fix the trained embedding vectors (corresponding to the row and column vectors in the table), and refine the allocation to minimize the training loss, which is a minimum weight perfect matching problem in graph theory and can be effectively solved. (3) We repeat the second step until certain stopping criterion is met.

We evaluate LightRNN using the language modeling task on several benchmark datasets. The experimental results show that LightRNN achieves comparable (if not better) accuracy to state-of-the-art language models in terms of perplexity, while reducing the model size by a factor of up to 100 and speeding up the training process by a factor of 2.

Please note that it is desirable to have a highly compact model (without accuracy drop). First, it makes it possible to put the RNN model into a GPU or even a mobile device. Second, if the training data is large and one needs to perform distributed data-parallel training, the communication cost for aggregating the models from local workers will be low. In this way, our approach makes previously expensive RNN algorithms very economical and scalable, and therefore has its profound impact on deep learning for NLP tasks.

## 2 Related work

In the literature of deep learning, there have been several works that try to resolve the problem caused by the large vocabulary of the text corpus.

Some works focus on reducing the computational complexity of the *softmax* operation on the output-embedding matrix. In [16, 17], a binary tree is used to represent a hierarchical clustering of words in the vocabulary. Each leaf node of the tree is associated with a word, and every word has a unique path from the root to the leaf where it is in. In this way, when calculating the probability of the next word, one can replace the original $|V|$-way normalization with a sequence of $\log|V|$ binary normalizations. In [9, 15], the words in the vocabulary are organized into a tree with two layers: the root node has roughly $\sqrt{|V|}$ intermediate nodes, each of which also has roughly $\sqrt{|V|}$ leaf nodes. Each intermediate node represents a cluster of words, and each leaf node represents a word in the cluster. To calculate the probability of the next word, one first calculates the probability of the cluster of the word and then the conditional probability of the word given its cluster. Besides, methods based

on sampling-based approximations intend to select randomly or heuristically a small subset of the output layer and estimate the gradient only from those samples, such as importance sampling [3] and BlackOut [12]. Although these methods can speed up the training process by means of efficient *softmax*, they do not reduce the size of the model.

Some other works focus on reducing the model size. Techniques [6, 21] like differentiated *softmax* and recurrent projection are employed to reduce the size of the output-embedding matrix. However, they only slightly compress the model, and the number of parameters is still in the same order of the vocabulary size. Character-level convolutional filters are used to shrink the size of the input-embedding matrix in [13]. However, it still suffers from the gigantic output-embedding matrix. Besides, these methods have not addressed the challenge of computational complexity caused by the time-consuming *softmax* operations.

As can be seen from the above introductions, no existing work has simultaneously achieved the significant reduction of both model size and computational complexity. This is exactly the problem that we will address in this paper.

# 3 LightRNN

In this section, we introduce our proposed LightRNN algorithm.

## 3.1 RNN Model with 2-Component Shared Embedding

A key technical innovation in the LightRNN algorithm is its 2-Component shared embedding for word representations. As shown in Figure 1, we allocate all the words in the vocabulary into a table. The $i$-th row of the table is associated with an embedding vector $x_i^r$ and the $j$-th column of the table is associated with an embedding vector $x_j^c$. Then a word in the $i$-th row and the $j$-th column is represented by two components: $x_i^r$ and $x_j^c$. By sharing the embedding vector among words in the same row (and also in the same column), for a vocabulary with $|V|$ words, we only need $2\sqrt{|V|}$ unique vectors for the input word embedding. It is the same case for the output word embedding.

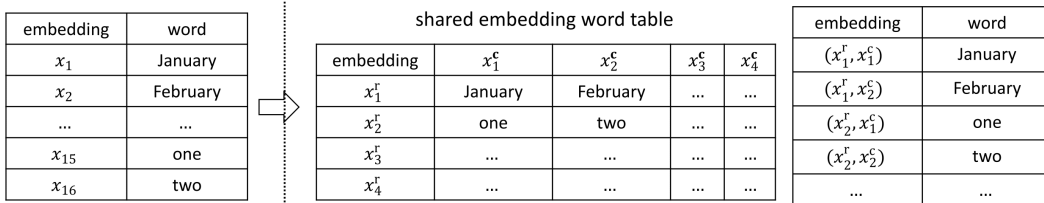

Figure 1: An example of the word table

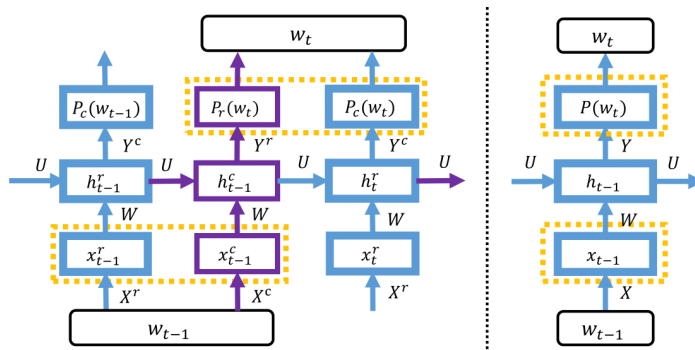

Figure 2: LightRNN (left) vs. Conventional RNN (right).

With the 2-Component shared embedding, we can construct the LightRNN model by doubling the basic units of a vanilla RNN model, as shown in Figure 2. Let $n$ and $m$ denote the dimension of a row/column input vector and that of a hidden state vector respectively. To compute the probability distribution of $w_t$, we need to use the column vector $x_{t-1}^c \in \mathbb{R}^n$, the row vector $x_t^r \in \mathbb{R}^n$, and the hidden state vector $h_{t-1}^r \in \mathbb{R}^m$.

The column and row vectors are from input-embedding matrices $X^c, X^r \in \mathbb{R}^{n \times \sqrt{|V|}}$ respectively. Next two hidden state vectors $h_{t-1}^c, h_t^r \in \mathbb{R}^m$ are produced by applying the following recursive

operations:

$$h_{t-1}^c = f(Wx_{t-1}^c + Uh_{t-1}^r + b) \quad h_t^r = f(Wx_t^r + Uh_{t-1}^c + b). \tag{1}$$

In the above function, $W \in \mathbb{R}^{m \times n}, U \in \mathbb{R}^{m \times m}, b \in \mathbb{R}^m$ are parameters of affine transformations, and $f$ is a nonlinear activation function (e.g., the sigmoid function).

The probability $P(w_t)$ of a word $w$ at position $t$ is determined by its row probability $P_r(w_t)$ and column probability $P_c(w_t)$:

$$P_r(w_t) = \frac{\exp(h_{t-1}^c \cdot y_{r(w)}^r)}{\sum_{i \in S_r} \exp(h_{t-1}^c \cdot y_i^r)} \qquad P_c(w_t) = \frac{\exp(h_t^r \cdot y_{c(w)}^c)}{\sum_{i \in S_c} \exp(h_t^r \cdot y_i^c)}, \tag{2}$$

$$P(w_t) = P_r(w_t) \cdot P_c(w_t), \tag{3}$$

where $r(w)$ is the row index of word $w$, $c(w)$ is its column index, $y_i^r \in \mathbb{R}^m$ is the $i$-th vector of $Y^r \in \mathbb{R}^{m \times \sqrt{|V|}}$, $y_i^c \in \mathbb{R}^m$ is the $i$-th vector of $Y^c \in \mathbb{R}^{m \times \sqrt{|V|}}$, and $S_r$ and $S_c$ denote the set of rows and columns of the word table respectively. Note that we do not see the $t$-th word before predicting it. In Figure 2, given the input column vector $x_{t-1}^c$ of the $(t-1)$-th word, we first infer the row probability $P_r(w_t)$ of the $t$-th word, and then choose the index of the row with the largest probability in $P_r(w_t)$ to look up the next input row vector $x_t^r$. Similarly, we can then infer the column probability $P_c(w_t)$ of the $t$-th word.

We can see that by using Eqn.(3), we effectively reduce the computation of the probability of the next word from a $|V|$-way normalization (in standard RNN models) to two $\sqrt{|V|}$-way normalizations. To better understand the reduction of the model size, we compare the key components in a vanilla RNN model and in our proposed LightRNN model by considering an example with embedding dimension $n = 1024$, hidden unit dimension $m = 1024$ and vocabulary size $|V| = 10M$. Suppose we use 32-bit floating point representation for each dimension. The total size of the two embedding matrices $X, Y$ is $(m \times |V| + n \times |V|) \times 4 = 80GB$ for the vanilla RNN model and that of the four embedding matrices $X^r, X^c, Y^r, Y^c$ in LightRNN is $2 \times (m \times \sqrt{|V|} + n \times \sqrt{|V|}) \times 4 \approx 50MB$. It is clear that LightRNN shrinks the model size by a significant factor so that it can be easily fit into the memory of a GPU device or a mobile device.

The cell of hidden state $h$ can be implemented by a LSTM [22] or a gated recurrent unit (GRU) [7], and our idea works with any kind of recurrent unit. Please note that in LightRNN, the input and output use different embedding matrices but they share the same word-allocation table.

### 3.2   Bootstrap for Word Allocation

The LightRNN algorithm described in the previous subsection assumes that there exists a word allocation table. It remains as a problem how to appropriately generate this table, i.e., how to allocate the words into appropriate columns and rows. In this subsection, we will discuss on this issue.

Specifically, we propose a bootstrap procedure to iteratively refine word allocation based on the learned word embedding in the LightRNN model:

(1)  For cold start, randomly allocate the words into the table.

(2)  Train the input/output embedding vectors in LightRNN based on the given allocation until convergence. Exit if a stopping criterion (e.g., training time, or perplexity for language modeling) is met, otherwise go to the next step.

(3)  Fixing the embedding vectors learned in the previous step, refine the allocation in the table, to minimize the loss function over all the words. Go to Step (2).

As can be seen above, the refinement of the word allocation table according to the learned embedding vectors is a key step in the bootstrap procedure. We will provide more details about it, by taking language modeling as an example.

The target in language modeling is to minimize the negative log-likelihood of the next word in a sequence, which is equivalent to optimizing the cross-entropy between the target probability distribution and the prediction given by the LightRNN model. Given a context with $T$ words, the

overall negative log-likelihood can be expressed as follows:

$$NLL = \sum_{t=1}^{T} -\log P(w_t) = \sum_{t=1}^{T} -\log P_r(w_t) - \log P_c(w_t). \tag{4}$$

$NLL$ can be expanded with respect to words: $NLL = \sum_{w=1}^{|V|} NLL_w$, where $NLL_w$ is the negative log-likelihood for a specific word $w$.

For ease of deduction, we rewrite $NLL_w$ as $l(w, r(w), c(w))$, where $(r(w), c(w))$ is the position of word $w$ in the word allocation table. In addition, we use $l_r(w, r(w))$ and $l_c(w, c(w))$ to represent the row component and column component of $l(w, r(w), c(w))$ (which we call row loss and column loss of word $w$ for ease of reference). The relationship between these quantities is

$$\begin{aligned} NLL_w &= \sum_{t \in S_w} -\log P(w_t) = l(w, r(w), c(w)) \\ &= \sum_{t \in S_w} -\log P_r(w_t) + \sum_{t \in S_w} -\log P_c(w_t) = l_r(w, r(w)) + l_c(w, c(w)), \end{aligned} \tag{5}$$

where $S_w$ is the set of all the positions for the word $w$ in the corpus.

Now we consider adjusting the allocation table to minimize the loss function $NLL$. For word $w$, suppose we plan to move it from the original cell $(r(w), c(w))$ to another cell $(i, j)$ in the table. Then we can calculate the row loss $l_r(w, i)$ if it is moved to row $i$ while its column and the allocation of all the other words remain unchanged. We can also calculate the column loss $l_c(w, j)$ in a similar way. Next we define the total loss of this move as $l(w, i, j)$ which is equal to $l_r(w, i) + l_c(w, j)$ according to Eqn.(5). The total cost of calculating all $l(w, i, j)$ is $\mathcal{O}(|V|^2)$, by assuming $l(w, i, j) = l_r(w, i) + l_c(w, j)$, since we only need to calculate the loss of each word allocated in every row and column separately. In fact, all $l_r(w, i)$ and $l_c(w, j)$ have already been calculated during the forward part of LightRNN training: to predict the next word we need to compute the scores (i.e., in Eqn.(2), $h_{t-1}^c \cdot y_i^r$ and $h_t^r \cdot y_i^c$ for all $i$) of all the words in the vocabulary for normalization and $l_r(w, i)$ is the sum of $-\log(\frac{\exp(h_{t-1}^c \cdot y_i^r)}{\sum_k (\exp(h_{t-1}^c \cdot y_k^r))})$ over all the appearances of word $w$ in the training data. After we calculate $l(w, i, j)$ for all possible $w, i, j$, we can write the reallocation problem as the following optimization problem:

$$\min_a \sum_{(w,i,j)} l(w, i, j) a(w, i, j) \quad \text{subject to}$$

$$\sum_{(i,j)} a(w, i, j) = 1 \quad \forall w \in V, \quad \sum_w a(w, i, j) = 1 \quad \forall i \in S_r, j \in S_c, \tag{6}$$

$$a(w, i, j) \in \{0, 1\}, \quad \forall w \in V, i \in S_r, j \in S_c,$$

where $a(w, i, j) = 1$ means allocating word $w$ to position $(i, j)$ of the table, and $S_r$ and $S_c$ denote the row set and column set of the table respectively.

By defining a weighted bipartite graph $\mathcal{G} = (\mathcal{V}, \mathcal{E})$ with $\mathcal{V} = (V, S_r \times S_c)$, in which the weight of the edge in $\mathcal{E}$ connecting a node $w \in V$ and node $(i, j) \in S_r \times S_c$ is $l(w, i, j)$, we will see that the above optimization problem is equivalent to a standard minimum weight perfect matching problem [18] on graph $\mathcal{G}$. This problem has been well studied in the literature, and one of the best practical algorithms for the problem is the minimum cost maximum flow (MCMF) algorithm [1], whose basic idea is shown in Figure 3. In Figure 3(a), we assign each edge connecting a word node $w$ and a position node $(i, j)$ with flow capacity 1 and cost $l(w, i, j)$. The remaining edges starting from source ($src$) or ending at destination ($dst$) are all with flow capacity 1 and cost 0. The thick solid lines in Figure 3(a) give an example of the optimal weighted matching solution, while Figure 3(b) illustrates how the allocation gets updated correspondingly. Since the computational complexity of MCMF is $\mathcal{O}(|V|^3)$, which is still costly for a large vocabulary, we alternatively leverage a linear time (with respect to $|\mathcal{E}|$) $\frac{1}{2}$-approximation algorithm [20] in our experiments whose computational complexity is $\mathcal{O}(|V|^2)$. When the number of tokens in the dataset is far larger than the size of the vocabulary (which is the common case), this complexity can be ignored as compared with the overall complexity of LightRNN training (which is around $\mathcal{O}(|V|KT)$, where $K$ is the number of epochs in the training process and $T$ is the total number of tokens in the training data).

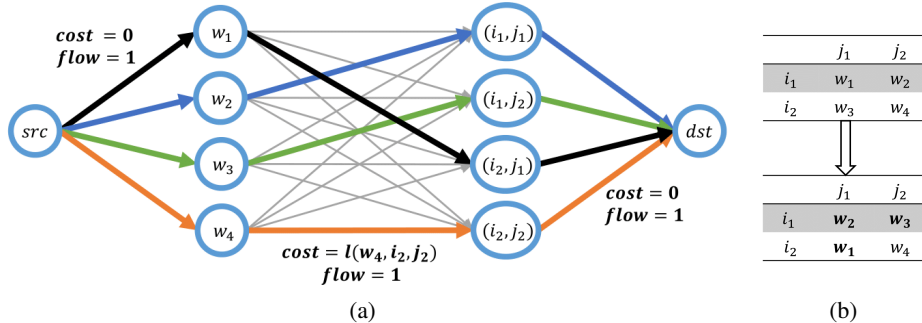

(a)                                                        (b)

Figure 3: The MCMF algorithm for minimum weight perfect matching

# 4  Experiments

To test LightRNN, we conducted a set of experiments on the language modeling task.

## 4.1  Settings

We use perplexity ($PPL$) as the measure to evaluate the performance of an algorithm for language modeling (the lower, the better), defined as $PPL = \exp(\frac{NLL}{T})$, where $T$ is the number of tokens in the test set. We used all the linguistic corpora from 2013 ACL Workshop Morphological Language Datasets (ACLW) [4] and the One-Billion-Word Benchmark Dataset (BillionW) [5] in our experiments. The detailed information of these public datasets is listed in Table 1.

Table 1: Statistics of the datasets

| Dataset | #Token | Vocabulary Size |
|---|---|---|
| ACLW-Spanish | 56M | 152K |
| ACLW-French | 57M | 137K |
| ACLW-English | 20M | 60K |
| ACLW-Czech | 17M | 206K |
| ACLW-German | 51M | 339K |
| ACLW-Russian | 25M | 497K |
| BillionW | 799M | 793K |

For the ACLW datasets, we kept all the training/validation/test sets exactly the same as those in [4, 13] by using their processed data [1]. For the BillionW dataset, since the data[2] are unprocessed, we processed the data according to the standard procedure as listed in [5]: We discarded all words with count below 3 and padded the sentence boundary markers <S>,<\S>. Words outside the vocabulary were mapped to the <UNK> token. Meanwhile, the partition of training/validation/test sets on BillionW was the same with public settings in [5] for fair comparisons.

We trained LSTM-based LightRNN using stochastic gradient descent with truncated backpropagation through time [10, 25]. The initial learning rate was 1.0 and then decreased by a ratio of 2 if the perplexity did not improve on the validation set after a certain number of mini batches. We clipped the gradients of the parameters such that their norms were bounded by 5.0. We further performed dropout with probability 0.5 [28]. All the training processes were conducted on one single GPU K20 with 5GB memory.

## 4.2  Results and Discussions

For the ACLW datasets, we mainly compared LightRNN with two state-of-the-art LSTM RNN algorithms in [13]: one utilizes hierarchical *softmax* for word prediction (denoted as HSM), and the other one utilizes hierarchical *softmax* as well as character-level convolutional filters for input embedding (denoted as C-HSM). We explored several choices of dimensions of shared embedding for LightRNN: 200, 600, and 1000. Note that 200 is exactly the word embedding size of HSM and C-HSM models used in [13]. Since our algorithm significantly reduces the model size, it allows us to use larger dimensions of embedding vectors while still keeping our model size very small. Therefore, we also tried 600 and 1000 in LightRNN, and the results are showed in Table 2. We can see that with larger embedding sizes, LightRNN achieves bet-

[1]https://www.dropbox.com/s/m83wwnlz3dw5zhk/large.zip?dl=0
[2]http://tiny.cc/1billionLM

Table 3: Runtime comparisons in order to achieve the HSMs' baseline $PPL$

| ACLW | | |
|---|---|---|
| Method | Runtime(hours) | Reallocation/Training |
| C-HSM[13] | 168 | – |
| LightRNN | 82 | 0.19% |
| BillionW | | |
| Method | Runtime(hours) | Reallocation/Training |
| HSM[6] | 168 | – |
| LightRNN | 70 | 2.36% |

Table 4: Results on BillionW dataset

| Method | $PPL$ | #param |
|---|---|---|
| KN[5] | 68 | 2G |
| HSM[6] | 85 | 1.6G |
| B-RNN[12] | 68 | 4.1G |
| LightRNN | **66** | **41M** |
| KN + HSM[6] | 56 | – |
| KN + B-RNN[12] | 47 | – |
| KN + LightRNN | **43** | – |

ter accuracy in terms of perplexity. With 1000-dimensional embedding, it achieves the best result while the total model size is still quite small. Thus, we set 1000 as the shared embedding size while comparing with baselines on all the ACLW datasets in the following experiments.

Table 2: Test $PPL$ of LightRNN on the ACLW-French dataset w.r.t. embedding sizes

| Embedding size | $PPL$ | #param |
|---|---|---|
| 200 | 340 | 0.9M |
| 600 | 208 | 7M |
| 1000 | **176** | 17M |

Table 5 shows the perplexity and model sizes in all the ACLW datasets. As can be seen, LightRNN significantly reduces the model size, while at the same time outperforms the baselines in terms of perplexity. Furthermore, while the model sizes of the baseline methods increase linearly with respect to the vocabulary size, the model size of LightRNN almost keeps constant on the ACLW datasets.

For the BillionW dataset, we mainly compared with BlackOut for RNN [12] (B-RNN) which achieves the state-of-the-art result by interpolating with KN (Kneser-Ney) 5-gram. Since the best single model reported in the paper is a 1-layer RNN with 2048-dimenional word embedding, we also used this embedding size for LightRNN. In addition, we compared with the HSM result reported in [6], which used 1024 dimensions for word embedding, but still has 40x more parameters than our model. For further comparisons, we also ensembled LightRNN with the KN 5-gram model. We utilized the KenLM Language Model Toolkit [3] to get the probability distribution from the KN model with the same vocabulary setting.

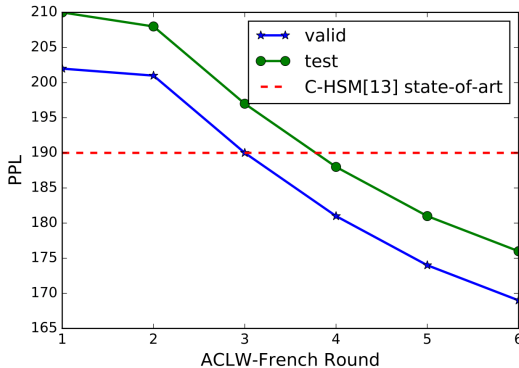

Figure 4: Perplexity curve on ACLW-French.

The results on BillionW are shown in Table 4. It is easy to see that LightRNN achieves the lowest perplexity whilst significantly reducing the model size. For example, it reduces the model size by a factor of 40 as compared to HSM and by a factor of 100 as compared to B-RNN. Furthermore, through ensemble with the KN 5-gram model, LightRNN achieves a perplexity of 43.

In our experiments, the overall training of LightRNN consisted of several rounds of word table refinement. In each round, the training stopped until the perplexity on the validation set converged. Figure 4 shows how the perplexity gets improved with respect to the table refinement on one of the ACLW datasets. Based on our observations, 3-4 rounds of refinements usually give satisfactory results.

Table 3 shows the training time of our algorithm in order to achieve the same perplexity as some baselines on the two datasets. As can be seen, LightRNN saves half of the runtime to achieve the same perplexity as C-HSM and HSM. This table also shows the time cost of word table refinement in the whole training process. Obviously, the word reallocation part accounts for very little fraction of the total training time.

Table 5: $PPL$ results in test set for various linguistic datasets on ACLW datasets. *Italic results* are the previous state-of-the-art. #P denotes the number of Parameters.

| | | | $PPL$ on ACLW test | | | |
|---|---|---|---|---|---|---|
| Method | Spanish/#P | French/#P | English/#P | Czech/#P | German/#P | Russian/#P |
| KN[4] | 219/– | 243/– | 291/– | 862/– | 463/– | 390/– |
| HSM[13] | 186/61M | 202/56M | 236/25M | 701/83M | 347/137M | 353/200M |
| *C-HSM*[13] | *169*/48M | *190*/44M | *216*/20M | *578*/64M | *305*/104M | *313*/152M |
| LightRNN | **157**/**18M** | **176**/**17M** | **191**/**17M** | **558**/**18M** | **281**/**18M** | **288**/**19M** |

Figure 5 shows a set of rows in the word allocation table on the BillionW dataset after several rounds of bootstrap. Surprisingly, our approach could automatically discover the semantic and syntactic relationship of words in natural languages. For example, the place names are allocated together in row 832; the expressions about the concept of time are allocated together in row 889; and URLs are allocated together in row 887. This automatically discovered semantic/syntactic relationship may explain why LightRNN, with such a small number of parameters, sometimes outperforms those baselines that assume all the words are independent of each other (i.e., embedding each word as an independent vector).

| row 832 | Karwan Narok Cocodrie Noja Anambra Alaska. Lantau Willmar Zululand Tianmen  … |
|---|---|
| row 852 | 281-211 3-6-0 17-of-44 21-for-27 100-64 1,173-767 10-to-2 7-and-5 15,350 of-15 … |
| row 861 | 103-run 12-way 23-hit 151-game 13-ball 105-meter 302-minute 189-yard 67-foot   … |
| row 872 | totaled hunted rigged scored vetoed inflicted froze swam won dried raged smiled … |
| row 877 | plods riles hankers misbehaves contrives utilizes disbands computes propagates … |
| row 887 | www.angiotech.com www.huntsman.com media.floridarealtors.org 2010.census.gov … |
| row 889 | years. decade evening hours. weeks spring summer. day-and-a-half April-to-June   … |
| row 891 | 44kg 63pc 170mph 18cm 22C 12A 150bp 17st 656ft 2Mbps 680g 10x 13ph. 2M  … |

Figure 5: Case study of word allocation table

## 5   Conclusion and future work

In this work, we have proposed a novel algorithm, LightRNN, for natural language processing tasks. Through the 2-Component shared embedding for word representations, LightRNN achieves high efficiency in terms of both model size and running time, especially for text corpora with large vocabularies.

There are many directions to explore in the future. First, we plan to apply LightRNN on even larger corpora, such as the ClueWeb dataset, for which conventional RNN models cannot be fit into a modern GPU. Second, we will apply LightRNN to other NLP tasks such as machine translation and question answering. Third, we will explore $k$-Component shared embedding ($k > 2$) and study the role of $k$ in the tradeoff between efficiency and effectiveness. Fourth, we are cleaning our codes and will release them soon through CNTK [27].

**Acknowledgments**

The authors would like to thank the anonymous reviewers for their critical and constructive comments and suggestions. This work was partially supported by the National Science Fund of China under Grant Nos. 91420201, 61472187, 61502235, 61233011 and 61373063, the Key Project of Chinese Ministry of Education under Grant No. 313030, the 973 Program No. 2014CB349303, and Program for Changjiang Scholars and Innovative Research Team in University. We also would like to thank Professor Xiaolin Hu from Department of Computer Science and Technology, Tsinghua National Laboratory for Information Science and Technology (TNList) for giving a lot of wonderful advices.

## Footnotes

[3] http://kheafield.com/code/kenlm/

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
