[Reviews · NeurIPS 2016]

Reviewer 1

Summary

The work describes a new method to improve a model's performance and significantly reduce the number of parameters by factorizing a vocabulary's embedding into two shared components (row and column). They show language modelling results with a newly proposed RNN algorithm on this factored embedding.

Qualitative Assessment

This work provides a novel and effective way to reduce the number of parameters for models that require handling of large vocabularies. The large drop in model size by several orders of magnitude could effectively allow some large models to be ported to the phone, which may not have been possible previously. I find it really interesting that a single method can improve both input parameter size and output size whereas previous work on softmaxes have only tackled the output side. However, I find that some technical details are lacking and the description can be confusing in some places. In particular, I find figure 2 and the unnumbered equation after Eq 1 confusing. For the 2C-RNN it seems that the output label w_t is dependent not only on the previous timestep x_{t-1}^c but also on itself: x_t^r (from w_t). In this case how does inference work? It's also unclear what the total cost of refining the word allocation is. The text seems to make no reference to the cost of calculating l(w, i, j) for all possible w, i, j or provide a way of efficiently calculating it other than doing forward prop for all possible combinations. The fact that the 2C-RNN now has 2x the number of timesteps as the original RNN could also be a problem, for example for the backprop of gradients. It would have been useful to see how this affects optimization and also how many words are moved at each round. I.e. are fewer words moved at each round? Table 3 and table 4 seem to show that much larger embeddings are required in the 2C-RNN case to achieve equivalent performance on the ACLW task. When the table states an embedding size of 1000 is that in total for a word or for a column/row? Since so much bigger embeddings are required, it seems that W would need to be much greater so slowing down inference time. As a result of a significantly larger W and 2x the number of processing steps (timesteps) the better perplexity results on ACLW are less surprising. The results on BillionW however are much more impressive for such a large reduction in the number of parameters. It would also have been nice to compare against quantization methods. Minor notes: L17: Verb tense: sacrifice L27: gate -> gating L38 and L41: M and GB should be in ordinary font, not italics. L42: It's also above the capability of most desktop machines, however, often models these large are distributed across multiple GPUs or machines. What makes it nice is that these new models can fit on a mobile device. L119: Next -> The next L125: S_r and S_c are not used until the next page. L161: This is a confusing way of describing it. L181: Dst is the usual way of denoting the destination rather than des. Figure 3: The bolded arrows should be colored too. L256: Too many 'the'. L259: Why stop training at this point instead of running to convergence?

Confidence in this Review

3-Expert (read the paper in detail, know the area, quite certain of my opinion)


Reviewer 2

Summary

Consider a word level language model. The majority of its parameters lie in the embedding matrix, as each word is typically assigned a 100-1000 dimensional vector. This submission presents a novel factorization of the embedding matrix, that quadratically reduces the memory requiremnts. In turn, this allows the use of larger word embeddings and leads to state-of-the-art performance on ACLW and Billion Words benchmarks. Traditionally, word embeddings are kept in a single table of size #words x embedding_dim. In this contribution, the authors create two small tables, both of size sqrt(#words) x embedding-dim, that are called the row-embedding matrix and the column-embedding matrix. Each word is assigned to an embedding from the row-embedding matrix and from the column-embedding one. Thus the embeddings of words are shared between words. However, this change substantially reduces the memory requirements. During model use, the RNN makes two steps for each word - it updates its hidden state first using the row-embedding matrix, then using the column-embedding matrix. Likewise, the probability of sampling a word is factorized into two probabilities: of the probability assigned to the word's position of the word in the row-matrix, and of of the probability assigned to the word's position in the column matrix. This solves another important problem, by reducing the amount of computation needed to compute the probability of an output word. The key ingredient of the proposed contribution is an algorithm that assigns word and column indexes to words, therefore establishing the sharing pattern for the embeddings. The authors frame it as a matching problem, and elegantly use a heuristic solution to the Minimum Cost Maximum Flow graph problem.

Qualitative Assessment

This paper solves two important practical problems of 1. reducing the size of the word embedding matrix and 2. reducing the computations of word probabilities, obviating the need for a hierarchical Softmax or Monte Carlo estimations of the model's training cost. The proposed model makes no assumptions about the structure of the words, which makes it potentially useful outside of NLP. In contrast, character-based word embedding models also reduce the model size, but do need to access the internal structure of words (i.e. their characters). The presentation is sufficiently clear, though I have the following questions: 1. from equation (1) and Fig. 2 it seems that the model sees the t-th word before predicting it. Still, there is an information bottleneck, as the word is aliased with sqrt(#words) that are assigned the same row-embedding. However, this is still surprising - could the author comment whether this is intentional, and how do the models behave when the word w_t is generated using information about words up to t-1 only? 2. how is the cost (4) computed? I guess that there are separate matrices for input-word embedding and output-word embedding, that share the word-allocation table. To compute eq. (4) is only the output allocation changed (which is fast to compute, as the hidden states of the RNN do not change and words are conditionally independent given the hidden states), or is the input allocation changed too? 3. how often in practice is the index reallocation (step 3) phase run?

Confidence in this Review

2-Confident (read it all; understood it all reasonably well)


Reviewer 3

Summary

In this paper, the authors try to simultaneously reduce both the model size and the computational complexity of RNN models with huge vocabulary sizes, via a 2-component shared embedding representation based on shared rows and columns of a table, leading to 2*sqrt(V) unique vectors instead of V. To learn the word allocation matrix, they use a bootstrap framework that iteratively refines the allocation based on loss minimization with a min-weight perfect matching problem. While the idea and motivation is useful, and the paper well-explained, the evaluation is weak to have strong takeaways of how well the technique works.

Qualitative Assessment

After author response: Thanks for answering some of my questions -- I updated some of my scores accordingly. However, I still encourage the authors to answer the rest of the questions, esp. the evaluation on the skipped datasets from Kim2015 C-HSM paper [13] and from Chen2015 [6], trying a downstream task with a classification metric (instead of ppl), and including clear timing/speed experiments. ------------------ Strengths: -- the motivation for reducing both model size and computation complexity is useful. -- the related work is quite well summarized and differentiated from. -- the 2-component idea is well-explained. -- the parameter reduction on the few experiments that the authors tried seems significant. Weaknesses: -- the evaluation is not strong enough to get any takeaways of whether this method scales or is useful for real tasks. -- first, the authors only report on a language modeling task and with a perplexity metric (which is just a function of the loss being minimized). They should have reported on at least one downstream task with classification metrics, even if a simple one like part-of-speech tagging. -- secondly, even on language modeling, the authors only report on a very small subset of experiments, whereas previous work they compare to has many more experiments. E.g., the Kim2015 C-HSM paper [13] reports on 6 languages for both data-small and data-large settings, and on English PTB, but here only two (French, Spanish) of those 13-14 experiments are reported on, without giving any reasons. Similarly, the Chen2015 [6] paper reports on Eng PTB, Gigaword, and BillionW. Btw, the state-of-the-art BillionW results are much stronger in [6], nearly 56-57 PPL. I understand that these are KN+Softmax combinations but that's simple enough for the authors to try too. -- the timing/speed experiments need more content/discussion in the paper. -- In NLP in general, most state of the art systems are happy with 100K-300K vocabulary sizes because all other rare words are backed off to some UNK vectors or character information. Hence, this paper should first show/motivate whether a very large vocab is even needed or not, and in cases where it is needed, how much does it help. Model questions: -- what happens for general k-component RNNs (at least a theoretical discussion) would be useful to add for completeness. -- In Eqn2, are there other interesting functions that can be tried instead of Pc*Pr?

Confidence in this Review

2-Confident (read it all; understood it all reasonably well)


Reviewer 4

Summary

This paper studies problems encountered in language modeling with neural network architectures. The goal is to reduce the model sizes of such architectures, which tend to have many parameters caused by the size of the vocabulary. To this end, the authors propose a method to reduce the size of both the input-embedding, and the output embedding. The goal of the proposed approach is different from methods such as importance-sampling that only reduce the size of the softmax but not the size of the input embedding layer. It is also different from methods that do character level input embeddings as they reduce the size of the input embedding matrix but not the size of the softmax layer. Instead of representing each word with a unique row or column in an embedding matrix which requires |V| vectors, the authors propose to represent each word with two vectors by placing each word within a slot in a table. Therefore, words in the same row share a vector, and words in the same column as share a vector. An iterative approach is used to refine allocation of words to table slots. The approach makes use of the minimum cost maximum flow (MCMF) algorithm. Experiments were done on three datasets: two smaller datasets, and one larger one with approximately 1 Billion tokens in its sequences.

Qualitative Assessment

** The paper’s main goal is to improve the efficiency of exiting models while maintaining performance levels, but only perplexity and the number of parameters of the models are reported. I would have liked to see runtimes, just like timing numbers were reported in the paper that introduced the 1 billion benchmark (Chelba et al 2013). ** The iterative approach for allocating words to table slots makes use of an algorithm with quadratic complexity in the size of the vocabulary. The authors state that this is a small fraction of the training time given the complexity of the RNN model, 2C-RNN. Given that the baselines do not have this table allocation component, it would make sense for the authors to report runtimes in addition to the number of parameters of the 2c-RNN ** Perplexity achieved on the billion word benchmark by the main model, 2C-RNN [bootstrap], is 66; while this is better than the baselines reported in the paper, the authors chose to not report an even lower perplexity of 51 that was reported in (Chelba et al 2013). It would be good to good to justify this decision. ** Overall, the proposal 2 component embedding approach is interesting. However, the reported numbers on the 1 billion word benchmark are worse than the best results reported in (Chelba et al 2013). In addition, the authors fail to report run times so we do not know how much additional training time is added by the table allocation optimizer.

Confidence in this Review

2-Confident (read it all; understood it all reasonably well)


Reviewer 5

Summary

This paper proposed a 2-Component shared embedding for word representations, and used a bootstrap procedure to iteratively learn the word embedding and word allocation. Experiments on ACLW and BillionW dataset shows that the proposed method significantly shrink the model size and speed up the training process without sacrifice of accuracy.

Qualitative Assessment

1: The proposed 2-Component shared embedding is a very interesting and neat idea. It would be great to see the performance on some real world applications and k-Component embedding. 2: One small drawback of the 2-Component embedding assumptions is that the similar word will have the same number (same number on each row). If the future method could automatically learn that number, maybe it can be better. 3: What is the approximate time for the refine the allocation step? this time should also count when comparing the speed for different methods.

Confidence in this Review

2-Confident (read it all; understood it all reasonably well)


Reviewer 6

Summary

The authors applied a cartesian product space on the word embeddings to address the high dimensionality problem in natural language tasks with RNNs. A 2-D look up table is used for encoding/decoding word vectors. A novel bootstrap step is proposed to iteratively learn a word allocation table based on minimum weighted bipartite matching. The model is tested on several language modeling tasks and achieved the state-of-the-art results, beating hierarchical softmax by large margin, while significantly reducing the number of parameters.

Qualitative Assessment

The paper is a natural extension of an 1-layer word embedding look-up table towards a 2-layer look-up table. The bootstrap step is promising since it learns a good word allocation (similar to word clustering). There could be a large impact on various natural language applications. I don't see why this work is titled with Recurrent Neural Network. One could use the 2-layer look-up table with a traditional RNN: just concatenate the word-vector from the both components, and split the output vector into two vectors. Can you explain the difference and compare performance of a traditional RNN, and an alternating 2-C RNN? If they are similar, then the real contribution is the design of a 2-layer input/output embedding. If this is true then maybe the 2-layer look-up table can work with other feed-forward models such as BOW, together with the bootstrap step, and see if it can beat the standard word embedding model. It would be nice to include some comparison in terms of model training/inference time consumption, and some plots of the learned word clusters.

Confidence in this Review

3-Expert (read the paper in detail, know the area, quite certain of my opinion)